# Plant attribute extraction: An enhancing three-stage deep learning model for relational triple extraction

Zhihao Zong[1], Hongtao Shan[1], Gaoyu Zhang [ID][2]*, George Xianzhi Yuan [ID][3,4], Shuyi Zhang[5]

**1** School of Electronic and Electrical Engineering, Shanghai University of Engineering Science, Shanghai, China, **2** School of Information Management, Shanghai Lixin University of Accounting and Finance, Shanghai, China, **3** College of Science, Chongqing University of Technology, Chongqing, China, **4** Business School, Chengdu University, Chengdu, China, **5** Zhong Qiao Vocational and Technical University, Shanghai, China

\* 20089840@lixin.edu.cn

## Abstract

Various plant attributes, such as growing environment, growth cycle, and ecological distribution, can provide support to fields like agricultural production and biodiversity. This information is widely dispersed in texts. Manual extraction of this information is highly inefficient due to a fact that it not only takes considerable time but also increases the likelihood of overlooking relevant details. To convert textual data into structured information, we extract relational triples in the form of (subject, relation, object), where the subject represents the names of plants, the object represents the plant attributes, and the relation represents the classification of plant attributes. To reduce complexity, we employ a joint extraction of entities and relations based on a tagging scheme. The task is broken down into three parts. Firstly, a matrix is used to simultaneously match plant entities and plant attributes. Then, the predefined categories of plant attributes are classified. Finally, the categories of plant attributes are matched with entity-attribute pairs. The tagging-based method typically utilizes parameter sharing to facilitate interaction between different tasks, but it can also lead to issues such as error amplification and instability in parameter updates. Thus, we adopt improved techniques at different stages to enhance the performance of our model. This includes adjustment to the word embedding layer of BERT and optimization in relation prediction. The modification of the word embedding layer is intended to better integrate contextual information during text representation and reduce the interference of erroneous information. The relation prediction part mainly involves multi-level information fusion of textual information, thereby making corrections and highlighting important information. We name the three-stage method as "Bwdgv". Compared to the currently advanced PRGC model, the F1-score of the proposed method has an improvement of 1.4%. With the help of extracted triples, we can construct knowledge graphs and other tasks to better apply various plant attributes.

**Data availability statement:** The datasets are available in the GitHub repository. the dataset can be accessed through the following link: https://github.com/zzhAB2/datasethttps://github.com/Robin-WZQ/CBLUE_CMeIE_model.

**Funding:** This work was supported by National Natural Science Foundation of China (Grant Number 71971031). The funder had no role in study design, data collection and analysis, decision to publish, or preparation of the manuscript.

**Competing interests:** The authors have declared that no competing interests exist.

# 1. Introduction

In nature, plants are essential for providing energy to other organisms and forming intricate food webs and ecosystems [1,2]. In human society, plants have multiple social values, including their aesthetic appeal in urban greening and cultural significance [3]. With continuous advancements in ecology and other scientific fields, more functions and information about plants are being discovered [4,5]. Effectively using this extensive plant information can promote their application in agriculture, urban greening and more [6,7].

Text serves as the primary medium for conveying and sharing this extensive plant information and manual extraction of specific information from it is a challenging and time-consuming process. Relational triple extraction is a crucial task in natural language processing, which simplifies complex information by extracting specific events and relations. Through relational triple extraction, we can efficiently complete data processing and utilization [8]. In addition, Relational triple extraction can be categorized into two types based on whether the set of relations is determined: open-domain relation extraction and restricted-domain relation extraction. Open-domain relation extraction is not limited to predefined relations and can be expanded to discover new relations. But in the context of botany, the extraction of information related to specific terms and relationships needs to be ensured for accuracy and reliability. All categories of extracted attributes should be applicable to all plant entities. Thus, in this endeavor, restricted-domain relation extraction approach is employed. Categories of attributes have been predefined.

Within the realm of relational triple extraction, there exist two technical approaches: pipeline extraction and joint extraction [9]. Pipeline extraction model disassembles extraction into two distinct sequential tasks. Firstly, the task of named entity recognition extracts the entities and then the task of relation classification predicts the connections between these extracted entities. However, due to the independence of these two tasks, the task of relation classification cannot correct errors produced by the task of named entity recognition. Consequently, the accuracy of relational triple extraction is diminished. Conversely, the joint extraction model presents an end-to-end solution, where entities and relations are learned simultaneously.

Joint extraction can be divided into span-based, table-filling, and tagging-based approaches according to decoding strategies [10]. Span-based models directly recognize all entities without considering label transformations, but they may suffer from noise issues if there are a large number of entities. Table-filling models construct a table for each relation, with each entry indicating whether an entity pair possesses that particular relation. The design and decoding of the table greatly impact the model's performance, and an abundance of relations can escalate the model's complexity. Tagging-based models typically employ binary tagging sequences, which recognize entities through start tags and end tags. Binary tagging sequences also known as pointer networks, will be explained in the section of related work. Recognizing tags is simple and has low complexity; however, when the sample distribution is uneven, data sparsity becomes noticeable. Most of the current tagging-based

approaches are based on parameter sharing. Based on this, we first decompose the task into three subtasks: the extraction of plant entities and plant attributes, the classification of plant attributes, and the matching of attribute categories with entity-attribute pairs and then improve the effectiveness of parameter sharing from the following perspectives:

1. Due to multi-task parameter sharing, an error at one position can propagate backward, amplifying the impact of the mistake.

2. Dividing the model into different subtasks may cause it to focus on fitting a single task, potentially neglecting the others.

3. Different subtasks may have varying objectives, but they share the same set of parameters, which can lead to instability in parameter updates.

To mitigate the negative impact of parameter sharing, the initial step is to refine the encoder. The textual data is vectorized through text representation. Historically, bag-of-words model and GloVe model [11,12] were employed for text representation. However, these models utilized static word vectors, thereby failing to address the issue of word polysemy. Later, ELMo dynamically adjusted word vectors by bidirectional structure [13]. BERT further developed this concept through transformer [14,15]. In this study, we combine BERT with whole word masking to capture semantic dependencies between words effectively [16]. It modifies the strategy for generating training samples by masking not only the target subword but also its related sibling words. Additionally, we add perturbations [17] to the word embedding layer of the encoder. These perturbations introduce additional randomness and noise, compelling the encoder to learn comprehensive features. Finally, the perturbations are updated along with the parameters, stabilizing the training process.

Next, we optimize the classification of plant attributes by adopting a multi-level information fusion strategy. The first method directly converts vectors into sentence vectors through pooling to capture global semantic information. The second method has two cases for chunking the vectors. One is to segment characters according to the way words are formed, while the other is to segment according to the size of window. Then, the chunked vectors are pooled. The results of these methods are fused to minimize errors in classification. After classification, we match the categories of attributes with entity-attribute pairs. In matching, an attention mechanism is used to dynamically adjust the model's focus on different information, ensuring that the model can balance different tasks.

Based on this, we review several models. The ETL model [18] and the CasRel model [19] both first extract the head entity, and then simultaneously extract the relation and the corresponding tail entity based on the head entity. However, the relations between the two entities in the ETL model is unique, which would result in missing triples with different relations. CasRel solves this issue by extracting the tail entity separately for each relation, but the number of relations is large. In comparison, our model optimizes the extracting of relations, namely the classification of plant attributes, thereby reducing the redundancy of relations. For the TPLink model [20], it uses many matrices and extracts the head and tail entities with different matrices, which dissociates the dependency relations between entities. On this basis, OneRel [21] proposes a method that uses a matrix for each relation and extracts triples from this matrix. However, compared to our model, OneRel has more relations, which can generate more negative samples.

Currently there are relatively few studies on relation extraction in the field of botany, we propose the Bwdgv model with the entry point of mitigating the negative impact of parameter sharing. The model achieves extraction results with an accuracy of 88.5%, a precision of 89.8%, and a F1-score of 89.2%. The innovations of this work contain at least the following points:

(1) We enhance the diversity of contextual information by whole-word masking, and then introduce perturbations in model's embedding layer to improve the model's robustness.

(2) We propose a strategy to optimize subtasks, which involves devising a multi-level information fusion method for attribute classification and dynamically balancing subtasks during prediction.

The paper is organized into five sections. Section 1 provides an introduction. Section 2 presents a review of the current literature on relation extraction, outlining the strengths and weaknesses of existing approaches. In Section 3, we provide a detailed description of the workflow and various subtasks of the Bwdgv model. Section 4 focuses on the analysis of experimental data to validate the effectiveness of our proposed model. Finally, in Section 5, we summarize the work and propose directions for future improvement.

Additionally, in order to make the article easier to narrate and understand, a table of symbols is presented in Table 1.

## 2. Related work

Extraction of entities and relations holds significant importance within the realm of natural language processing. In previous approaches, entities and relations are often handled as separate tasks. For instance, Zelenko et al. [22] initially focus on extracting entities from text and then classify the relations between them. While this pipeline model serves its purpose, it is prone to error transfer, whereby mistakes made in entity extraction could affect the accuracy of relation classification.

To address this issue, numerous models for joint extraction of entities and relations have emerged. Ren et al. [23] propose a joint extraction model that relied on a priori features. Building upon this, Miwa and Bansal [24] introduce a parameter-sharing technique that enables entities and relations to be represented by the same encoder, facilitating multi-task learning.

### 2.1. Tagging-based joint extraction

The tagging-based approach is a decoding strategy of joint extraction that designs different tags to reasonably combine entity and relation tasks. Zheng et al. [25] introduce a sequence labeling scheme that includes information on both entities and relations, allowing for the mapping of entities and relations within a unified framework for joint decoding. The scheme directly extracts triples, where each entity is labeled with both a number representing the entity and the type of relations. However, the relations within each dataset are different, and the symbols for these relations need to be redefined in different domains. More importantly, the labeling scheme assumes that each entity has only one relation, making it impossible to identify entities with multiple relations. Katiyar et al. [26] split entities into multiple words and tag each word. However, operating on each token individually does not fully utilize entity boundary information, leading to dispersed information. Subsequently, in 2020, the CasRel model [19] improves extraction performance by incorporating pointer networks. The

**Table 1. Explanation of symbols.**

| Serial number | Symbol | Description |
|---|---|---|
| 1 | $p_i^{start\_s}, p_i^{end\_s}$ | The prediction results of the start and end positions of entities in a pointer network. |
| 2 | $x_i$ | Input vector of a pointer network. |
| 3 | $T(S)$ | Relational triple extracted from a sentence. |
| 4 | $h, t$ | Head entity and tail entity in a relational triple. |
| 5 | $r$ | Relation between the head entity and the tail entity. |
| 6 | $n, n_r, n_r^{pot}$ | The total number of entities, the number of potential relations and the number of actual relations. |
| 7 | $p_{rel}$ | The probability of predicted relations. |
| 8 | $x, \Delta x$ | The input vector of the model and perturbation. |
| 9 | $g$ | The derivative of loss at the word embeddings. |
| 10 | $e, e^{avg}$ | The input of predicting relations and the result of the operation of average pooling. |
| 11 | $p_{i,j}^{sub}, p_{i,j}^{obj}$ | Prediction of {B,I,O} for the current entity in sequence labeling. |
| 12 | $p_{i_{sub},\ j_{obj}}$ | The value of the element in global matrix. |

two classic tagging-based techniques are ETL and CasRel. ETL introduces two distinct strategies to handle entities and relations. In ETL framework, the task is divided into two interconnected subtasks: Head-Entity Extraction (HE) and Tail-Entity and Relation (TER). HE aims to identify the head entities, while TER focuses on identifying tail entities and relations associated with the head entities.

Although ELT employs a logical decomposition strategy to capture the semantic dependencies, it only provides a single recognition result for each entity, leading to omissions when extracting triples for the same entities. Similar to the concept of ELT, CasRel initially extracts head entities, followed by tail entities and relations. However, CasRel integrates pointer networks into the prediction process to tackle the aforementioned challenge. Fig 1 illustrates the architecture of the pointer network model.

The pointer network module is used to identify possible entities in the text. It utilizes two distinct binary classifiers to assign a binary label (0/1) to each token. Binary labels serve as indicators for identifying the boundaries of entities. The tag 0 indicates that the current token is not an entity to be recognized, while the tag 1 indicates that the current token is the start or end position of an entity. Equation (1) shows how to determine an entity's start and end positions:

$$p_i^{start_s} = \sigma\left(w_{start}x_i + b_{start}\right)$$
$$p_i^{end\_s} = \sigma\left(w_{end}x_i + b_{end}\right)$$

(1)

where variables $w_{sart}$, $b_{start}$, $w_{end}$, and $b_{end}$ are model parameters, $p_i^{start\_s}$ and $p_i^{end\_s}$ represent the probabilities of the entity's start and end positions, respectively. When recognizing entities, it's common to encounter overlapping entities, where the start positions are the same but the end positions are different. CasRel adopts the proximity matching method, which selects the end label closest to the start position of the overlapping entities as the end position. This method improves the performance of relation extraction, but it also has several limitations. Firstly, the training efficiency is suboptimal. After recognizing entities, CasRel randomly selects one entity for relation prediction, which limits the batch size to be set to one. Secondly, although the pointer network can handle overlapping triples, it generates many zeros, making matrices sparse.

Currently, sequence labeling and pointer network are widely used decoding schemes that can be applied to most scenarios. Thus, a sequence labeling approach is adopted in our research, but types of relations are removed to make it applicable to different kinds of texts. The tagging-based method is essentially multi-task learning. It not only uses different decoding methods but also shares the same encoder parameters in the extraction of entities and relations through parameter sharing. Zhe et al. improve the step of parameter sharing by re-modeling relations as mappings between entity pairs, thereby adding the interaction of relations [19]. Although the model employs parameter sharing method, there is an issue of exposure bias, meaning that the order of entity and relation tasks can lead to inconsistency in the model's parameters during training and inference. Yucheng et al. introduce a concept to address this issue by simultaneously handling the entity tasks and relation tasks [20]. Later, Heng et al. refine the tasks to reduce the complexity of individual tasks, thereby enhancing the performance of parameter sharing [27]. Based on the approaches mentioned above, we divide the task into

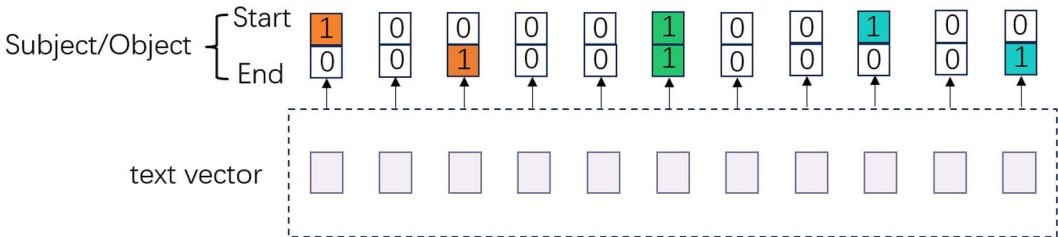

**Fig 1. Illustration of pointer network.**

different subtasks, and simultaneously extract relations and entities. Next, we make two adjustments. Adjustment to the word embeddings enhances text representation, enabling the model to balance different tasks and stabilize parameters. Adjustment to relation task improves the accuracy of relation extraction, which can better interact with other tasks and reduce the accumulation of errors from one stage to subsequent tasks.

## 2.2. Relation extraction in plant texts

Relation extraction based on deep learning has achieved great result. In plant texts, relation extraction is also a significant research direction, with extensive application prospects in assisting botanical research and other fields. Chen et al. [28] examine how specific landscape practices, leave a lasting imprint on urban flora, identifying 211 spontaneous plant species in 169 genera from 82 families. Baeksoo Kim et al. [29] develop a corpus focusing on plants and their treatment of diseases, expanding the application range of plant data. Cho et al. [30] extract plant entities based on word embeddings and Dalvi et al. [31] develop a robust deep CNN model to identify plant entities, enhancing the accuracy of the model in identifying medicinal plants related to skin health, specifically their leaves, trunks, and seeds. But they do not take into account the impact of relations. Zhang et al. [32] employ an approach combining BiLSTM and CRF to extract entities and relations, but the accuracy is suboptimal. Tang et al. [33] propose an entity recognition model and a relation extraction model, which are embedded with ALBERT, solving the problem of having poor ability of context fusion and computational efficiency. However, this method falls under the category of pipeline extraction models. Pipeline models are easier to implement but overlook the dependencies between entities and relations. Later, Zhu et al. [34] propose a scalable natural language annotation framework. The framework is a joint extraction model that employs convolutional neural networks to extract features at different depths and uses inter layer feature fusion to enhance feature representation capabilities. Compared to pipeline models, joint extraction models simultaneously extract entities and relations, naturally leveraging their interdependencies. In this paper, we will continue to explore the potential of joint extraction models in plant texts scenarios, improving the performance of joint extraction models.

## 3. The framework of the bwdgv model

The goal of the extraction of entities and relations is to extract triples. The extraction process can be represented as: $T(S) = \{(h, r, t)|, h, t \in E, r \in R\}$, where $S$ is the input sentence, $E$ is the set of entities, and $R$ is the set of predefined relations. $E$ contains the head entity $h$ and the tail entity $t$. The head entity and tail entity can also be referred to as the subject and object, respectively. Identifying entity pairs $(h, t)$ means finding the start and end positions of each element. Entity pairs and relations can be extracted simultaneously.

The task of extracting relational triples is divided into three subtasks: extraction of plant entities and plant attributes, classification of plant attributes, and matching of attribute categories with entity-attribute pairs. For given plant-related sentences, the plant entity is $h$, the plant attribute is $t$, and the attribute category is $r$. The first task extracts h and $t$, the second task extracts the potential $r$ and the third task selects the appropriate $r$ to match with $h$ and $t$. There are interactions between the subtasks. To alleviate issues such as error propagation and task imbalance, we first optimize the encoder. As shown in Fig 2:

We use BERT with whole word masking (BERT-wwm) to effectively capture the semantic dependencies between words [16]. Additionally, randomness and noise [17] are introduced in the embedding layer of the encoder, forcing the encoder to learn comprehensive features. Then, we optimize the classification of plant attributes by adopting a multi-level information fusion strategy. The first method directly converts vectors into sentence vectors through pooling. The second method involves two scenarios for segmenting vectors. One scenario is to segment based on the composition of words, and the other is to segment based on window size. The segmented vectors are then subjected to pooling. The results of these methods are fused to minimize errors in classification. After classification, $r$ is used to match with h and $t$. In this model, the total loss calculated at the end is the sum of the losses from the three subtasks, and perturbations need to be incorporated during parameter updates. As shown in Equation (2):

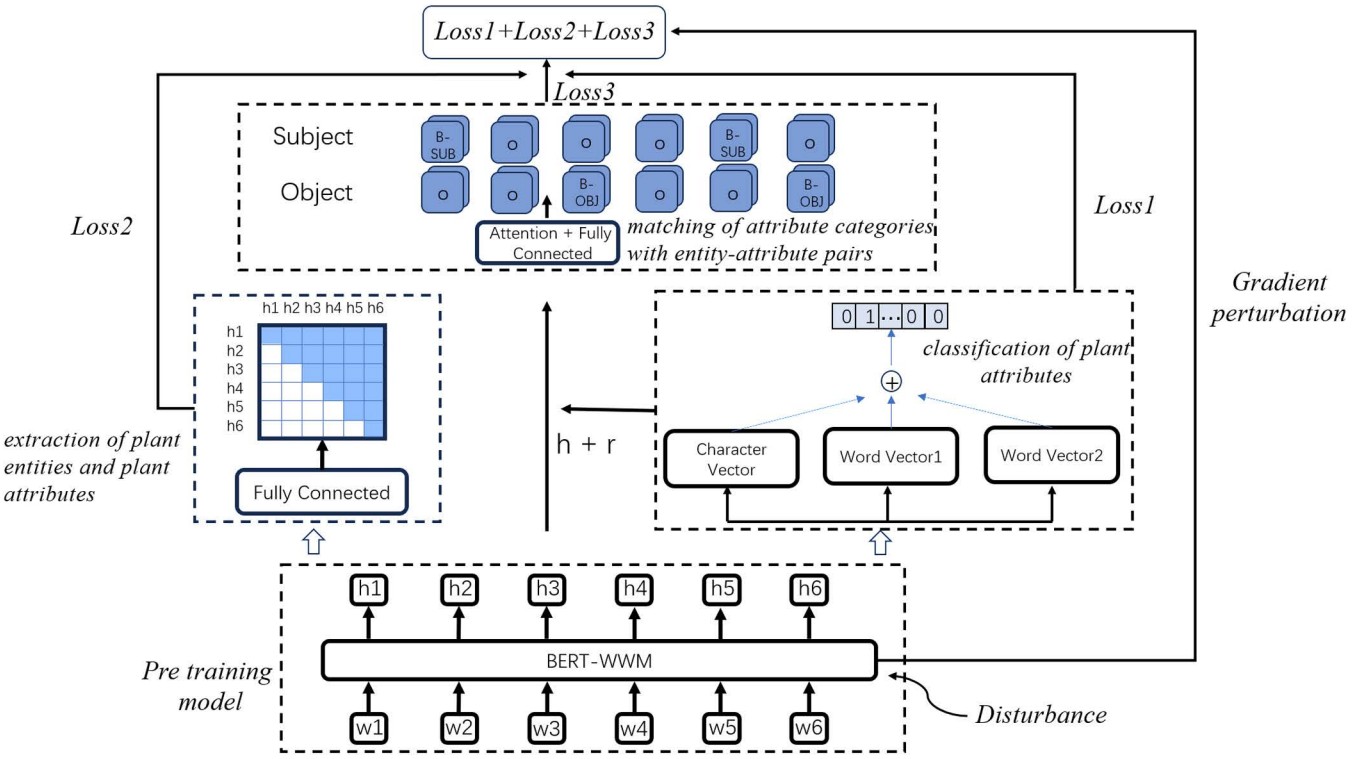

**Fig 2. The architecture of the model.**

$$loss1 = -\frac{1}{n}\sum_{i=1}^{nr}\left(y_i \log p_{rel} + (1-y_i)\log(1-p_{rel})\right)$$

$$loss2 = -\frac{1}{n^2}\sum_{i=1}^{n}\sum_{j=1}^{n}\left(y_i \log p_{i_{sub}j_{obj}} + (1-y_i)log(1-p_{rel})\right)$$

$$loss3 = -\frac{1}{2\times n\times n_r^{pot}}\sum_{a\in\{sub,obj\}}\sum_{j=1}^{n_r^{pot}}\sum_{i=1}^{n} y_{i,j}^a \log p_{i,j}^a$$

$$loss_{total} = loss1 + loss2 + loss3 \qquad (2)$$

where $n$ represents the number of head entities and tail entities, $n_r$ represents the number of all potential relations, and $n_r^{pot}$ represents the number of actual relations. The variable $a$ determines whether the loss calculation is performed for the head entity or the tail entity. $Loss1$ is the loss computation for the relations, $Loss2$ is the loss computation for the entities, and $Loss3$ is the loss computation for matching entity pairs with relations. Computing the loss for each subtask, a logarithmic function is utilized. The three sub-losses are combined in equal proportions.

### 3.1. Pre-trained encoder

Although parameter sharing helps to reduce the number of parameters in the model, it may also lead to a decrease in the model's ability to distinguish between errors. To mitigate the negative impact of parameter sharing, the following operations are employed. Firstly, BERT-wwm is used to more accurately represent text information, reducing the occurrence of errors. Secondly, perturbations are added to promote the model to learn more general features. Due to parameter sharing,

perturbations can be more effectively propagated to other layers, forcing the model to learn subtle changes in the data. Consequently, the model can enhance its adaptability to changes in input, reducing the propagation of errors.

In recent years, the BERT model has been the most commonly used pre-trained model. In Chinese datasets, characters are the basic units. Learning representations based on external information of characters easily overlooks the semantics of words. Therefore, we use BERT as the base model and choose the method of whole word masking (BERT-wwm).

To differentiate sentences, BERT adds the [CLS] token at the beginning of a sentence and the [SEP] token at the end of a sentence. These special tokens, [CLS] and [SEP], provide clear sentence boundaries and separation cues for entity pairs. Previous pre-trained models could only capture text information in one direction, limiting the model's representation ability. BERT is trained based on a bidirectional transformer architecture and combines word vectors, segment vectors, and position vectors. The word embedding layer enables the encoding to contain rich features. After adding the tokens, the sentence is transformed into vector $E$ through the word embedding layer. It can be formulated as: $E = E_C + E_S + E_P$, where $E_C$ is word vectors, $E_S$ is segment vectors, $E_P$ is position vectors. $E_C$ can be obtained by looking up the word embedding table, $E_S$ segments the sentence and assigns number, and $E_P$ represents the positional information of the words.

BERT is a masked language model (MLM). It masks part of the input content and restores the masked content when prediction. In training, BERT randomly selects tokens to replace with a probability of 15%. Each selected token has an 80% chance of being replaced with the [MASK] token, a 10% chance of being replaced with another token, and a 10% chance of remaining unchanged. However, it views each character as an independent unit, capturing external information of characters but ignoring the semantic information of words. To minimize the loss of semantic information, we use a whole word masking (wwm) technique. Bert-wwm employs whole word masking to simultaneously mask multiple tokens. We use Chinese datasets. For example,the input sentence is: "野鸢尾是多年生草本植物。" The differences in processing between Chinese and English are shown in Table 2.

In Chinese language, text is segmented by characters. For example, '野鸢尾' is split into '野', '鸢' and '尾'. In English, tools like wordpiece tokenizers split a word into smaller sub-words. For example, "Iris dichotoma" is split into 'Iris', 'di', '##ch', etc. Although the method of word segmentation differs between Chinese and English, the masking is the same. For the smallest unit of tokenization, the original masking randomly masks a unit, whereas the whole word masking masks the smallest unit of semantics, such as "野鸢尾", which would be entirely replaced by [MASK]. In English, words can be split into smaller sub-words, increasing the number of masks and the difficulty of prediction. In contrast, the whole word masking is more suitable for Chinese. The combination of characters forms more complex semantic units. The whole word masking improves the model's ability to handle long-range dependencies, allowing text representations to encompass more semantic information.

The text representations are used as input for subsequent subtasks. Considering the accuracy of the information and the stability of the training process, perturbations are introduced into the model. Perturbations can be implemented through data augmentation, but text data does not have operations like image flipping or cropping. The vectors converted from text are essentially one-hot vectors. A one-hot vector has a value of 1 at only one position and 0 everywhere else. The distance between the any two vectors is always $\sqrt{2}$. Therefore, adding perturbations to text vectors also has no

**Table 2. Differences with masking strategies in Chinese and English text.**

|  | Chinese | English |
|---|---|---|
| Original Sentence | 野鸢尾是多年生草本植物。 | Iris dichotoma is a perennial herb. |
| Tokenizer | 野 鸢 尾 是 多 年 生 草 本 植 物。 | Iris di ##ch ##oto ##ma is a per ##ennial herb. |
| Original Masking | 野 鸢 [M] 是 多 年 生 草 [M] 植 物。 | Iris di ##ch ##oto [M] is a per [M] herb. |
| Whole word masking | [M] [M] [M] 是 多 年 生 [M] [M] 植 物。 | [M] [M] [M] [M] [M] is a [M] [M] herb. |

effect. Manually adding characters is a way to introduce perturbations, but this method is inefficient and prone to semantic ambiguity. According to the improvements made by Giannis et al. on the LSTM model [35], fast gradient method (an adversarial training method) is adopted to construct perturbations. In the field of deep learning, the term "adversarial" typically refers to the generation of adversarial networks and adversarial samples [36]. Adding perturbations is equivalent to constructing adversarial samples and the procedure of identifying adversarial samples is commonly referred to as adversarial training. Adversarial training serves as an effective defense mechanism in the realm of adversarial attacks. This defensive approach is guided by Equation (3):

$$\min_\theta E(x, y) \sim D \left[ \max_{\Delta x \in \Omega} L(x + \Delta x, y; \theta) \right] \tag{3}$$

where $D$ is the training set, $x$ is the input, $y$ is the predicted label, $\theta$ is the model parameter, $\Delta x$ is the perturbation, $L$ is the loss function and $E$ is the function of empirical risk. In training, $x + \Delta x$ is the input with added perturbations. The training objective is to maximize the $L$, finding the worst-case perturbations as much as possible. $E$ is the output, and the goal is to minimize the external empirical risk, which means minimizing the model's error on the training data. Intuitively, the input should be as different from the original as possible, meaning the difference between the two inputs should be large; however, the output should be correctly identified, meaning the difference between the two outputs should be small. This formula points out the characteristics that adversarial examples should satisfy, as well as the method for generating adversarial examples (adversarial examples are a combination of the original sample $x$ and the added $\Delta x$ in some manner).

The core of adversarial training is to find an appropriate $\Delta x$. Word embedding layer, as the direct interface between the model and input text, serves to map words into dense vectors. Adding perturbations here can directly affect the model's understanding of vocabulary. In addition, compared to adding perturbations in deeper layers, perturbations in the word embedding layer are easier to control and do not have impact on the other structures of the model. Thus, we add perturbations in the word embedding layer. As shown in Fig 3.

The perturbation $\Delta x$ is directly added to the word embedding matrix. The input is first converted into a one-hot vector, and then transformed into a text vector through the word embedding layer. At this point, different samples will share the same perturbations, making the implementation easier. Finding the location where the perturbations need to be added, and then the perturbations can be calculated. The goal of perturbations $\Delta x$ is to maximize $L(x + \Delta x, y; \theta)$, and the direction of gradient ascent for $L$ is exactly the direction in which $L$ increases. We view the gradient of $L$ to be proportional to $\Delta x$. The calculation of the perturbations is shown in Equation (4):

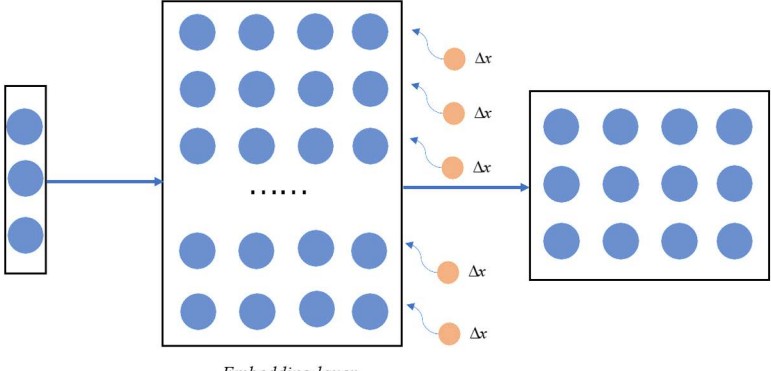

*Embedding layer*

**Fig 3. Word Embedding layer with perturbations.**

$$g = \nabla xL(x + \Delta x, y; \theta)$$
$$\Delta x = \varepsilon \frac{g}{\|g\|^2}$$

(4)

where g is the derivative of $L(x + \Delta x, y; \theta)$ at the word embeddings, $\varepsilon$ is the proportional coefficient, normalization is used to prevent perturbations from becoming too large. When constructing perturbations, the first step is to compute the loss of forward propagation and record the gradients of backward propagation. At this point, gradients obtained from word embeddings are regularized to obtain perturbations. And then, the perturbations are added to the word embedding to construct a new input $x + \Delta x$. Finally, the loss of forward propagation and the gradient of backward propagation are calculated once again. It should be noted that in the first training, the gradient does not need to be updated. In the second training, the gradient to be updated is the sum of the gradients calculated in the first and second computations. At this time, the model learns the features of both normal samples and adversarial samples simultaneously, better adapting to data changes and reducing fluctuation in the training. When updating the gradient, the input of word embeddings should revert to the original $x$. As the update without considering disturbances, the model reduces its reliance on adversarial samples, ensuring the stability of parameters. Based on this, the model learns how to resist noise at the input level and ensures the stability of the training process.

### 3.2. Classification of plant attributes

Errors can accumulate continuously due to the interaction between subtasks. We optimize the relation classification to minimize the errors from intermediate tasks. Relation classification is a multi-label classification task. Each sample can be classified into different labels, and these labels are not mutually exclusive. For example, saffron can have both a shape label and a cultural symbolic label. We convert the input text into sentence vectors and then use the sigmoid function for classification. When converting vectors, a multi-level semantic fusion strategy is employed. As is shown in Fig 4:

where, the input text is transformed by BERT into vector $e$ consisting of $n$ tokens. The input $e$ for relation classification is a three-dimensional vector. Assuming that batch_size = 3, sequence_length = 6, dim = 5. $e$ is processed into three vectors: $e_1$, $e_2$, $e_3$. $e_1$ is the original embedding vector $e$, which is transformed into a sentence vector directly through average

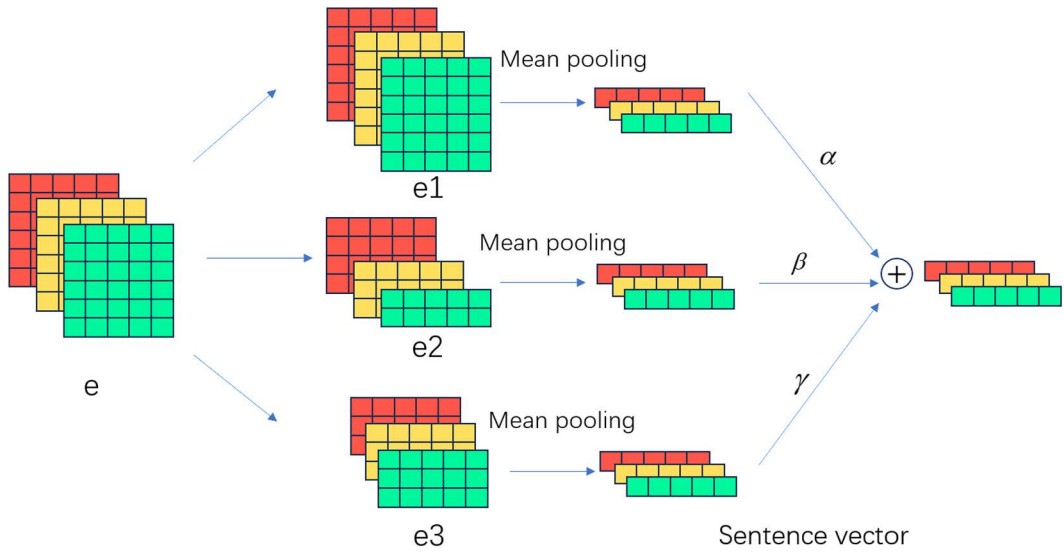

**Fig 4. Sentence vector based on multi-level semantic fusion.**

pooling. The output sentence vector at this moment is two-dimensional because the second dimension of $e_1$ has been averaged. Direct pooling for $e_1$ effectively captures global semantic information.

Chinese is segmented by characters, but different words contain varying numbers of characters. Based on the composition of words, $e_2$ can be obtained by combining different characters. We use jieba to segment Chinese text, and then mark the positions corresponding to the segmented words. Sentence vectors can be obtained by summing the characters at corresponding positions and then averaging. Since the number of results after summing the characters is different, the length of sequence_len, which is the second dimension of $e_2$, is inconsistent. $e_3$ can be obtained by segmenting characters based on a window size. We define the window size as 4. For every four characters, the vectors are summed and then averaged. Pooling is performed separately on $e_2$ and $e_3$ to obtain sentence vectors. We assign different weights to the three sentence vectors and combine them by summation. Then the combined vector is used for multi-label classification, as described in Equation (5):

$$\begin{aligned} e_1^{avg} &= Avgpool\left(e_1\right) \\ e_2^{avg} &= Avgpool\left(e_2\right) \\ e_3^{avg} &= Avgpool\left(e_3\right) \\ p_{rel} &= \sigma(w_r\left(\alpha e_1^{avg} + \beta e_2^{avg} + \gamma e_3^{avg}\right) + b_r) \end{aligned}$$

(5)

where $Avgpool$ is the average pooling, which converts vectors into sentence vectors, $\sigma$ is the sigmoid function, $w_r$ is the trainable parameter, $\alpha, \beta, \gamma$ are scaling factors used to weight different feature vectors. In prediction, we set a threshold $\lambda$. If the probability $p_{rel}$ exceeds $\lambda$, the current relation label is set to 1; otherwise, it is set to 0. The classification task takes into account different semantic relationships as well as the balance between global and local information.

### 3.3. Extraction of plant entities and plant attributes

The extraction of plant entities and plant attributes along with the classification of plant attributes can be performed simultaneously. As shown in Fig 1, we use a global matrix to predict plant entities(subjects) and plant attributes(objects) in the phase of entity pair extraction. The dimension of the global matrix is determined by the number of tokens in the sentence. If there are $n$ tokens, the shape of the matrix is $R^{n \times n}$. The columns of the global matrix represent the starting positions of the subjects, and the rows represent the starting positions of the objects. The elements in the global matrix indicate whether the subjects and objects begin from these positions. To obtain the global matrix, the tokens of the subjects and objects are first concatenated vertically. Then, a fully connected layer is used to compute the concatenated information. The value of each element in the global matrix represents the confidence between a subject and an object. The higher the confidence, the more likely the begin position will be predicted. The extraction of plant entities and plant attributes remains a classification task, thus we use the sigmoid function to predict the begin positions of subjects and objects. The resulting prediction is the global matrix, which can be expressed as Equation (6):

$$p_{i_{sub} j_{obj}} = \sigma\left(w_g\left[h_i^{sub}; h_j^{obj}\right] + b_g\right)$$

(6)

where, $p_{i_{sub}, j_{obj}}$ represents the value of the element in global matrix, $h_i^{sub}$ and $h_j^{obj}$ are the encoded representations of the starting positions of the subject and object, $w_g$ and $b_g$ are the training parameters, and $\sigma$ is the sigmoid function. Additionally, in the global matrix, we add a padding mask to mask out redundant entity information in prediction.

### 3.4. Matching of attribute categories with entity-attribute pairs

There are many categories of plant attributes. For each category, two sequence tagging operations are used to extract subjects and objects, respectively. Ensuring that the model can dynamically adjust its focus on different information, we introduce an attention mechanism for integrating the tasks. The model for extracting subjects and objects is shown in Fig 5:

where the text vector and relation vector are directly concatenated to form a new composite vector. This operation allows the model to consider both the overall text information and specific relational information, thereby enhancing the accuracy of subject and object identification. Next we use the attention mechanism to allocate weights to information, dynamically adjusting the model's focus on different pieces of information. The self-attention mechanism is adopted as the input consisted only of concatenated composite vector. For this composite vector, we use different linear transformations to generate the Query vector, Key vector, and Value vector. These transformations are usually fully connected layers, which map the input to different dimensional spaces. By computing the dot product between the Query vector and Key vector, a score matrix is obtained, reflecting the relevance of each element to all others. Then, to characterize the importance of the current information, the softmax function is used to convert the score matrix into a weight matrix. The weight matrix is finally used to perform a weighted calculation with the Value vector, resulting in the output vector for the position. After these steps, the text vector and the relation vector can be better integrated to highlight important information.

After assigning weights, we use a linear layer to expand the dimensions and then the BIO tag scheme is employed for prediction. BIO is a tagging scheme used for identifying positions, allowing for recognition of multiple entities. B indicates the beginning position of an entity, I indicates that the current position is inside the entity, and O indicates that the position is outside the entity. The BIO tagging scheme makes predictions for the subject and the object separately. The specific process is as shown in Equation (7):

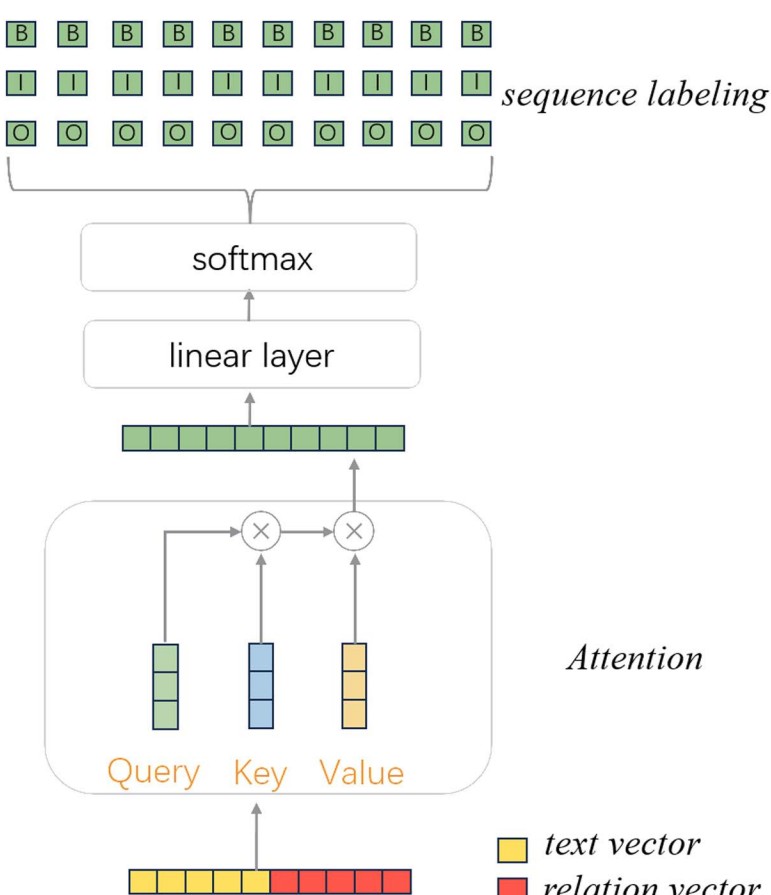

**Fig 5. Sequence labeling for entity pairs and relations.**

$$X = h_i + r_j$$
$$Q = XW_Q$$
$$K = XW_K$$
$$V = Xw_V$$
$$H = Softmax\left(\frac{QK^T}{\sqrt{d_k}}\right)V$$
$$p_{i,j}^{sub} = softmax\left(W_{sub}H + b_{sub}\right)$$
$$p_{i,j}^{obj} = softmax(W_{obj}H + b_{obj})$$

(7)

where, $W_Q$, $W_K$ and $W_V$ are matrices that transform vectors into different spaces, $h_i$ is the text vector, $r_j$ is the relation vector, $W_{sub}$, $W_{obj}$, $b_{sub}$, $b_{obj}$ are trainable parameters, $p_{i,j}^{sub}$ is the probability of subject and $p_{i,j}^{obj}$ is the probability of object. Concatenating $h_i$ and $r_j$, a new vector $X$ is obtained. The matrices $W_Q$, $W_K$ and $W_V$ project $X$ into a new space, then different attention weights are computed to emphasize important information. Finally, the weighted information is fed into a fully connected layer for prediction. After sequence tagging, all possible subjects and objects related to the relations are obtained. At this point, the scores of the corresponding subjects and objects are found in the global matrix. Subjects and objects with low scores are filtered out.

## 4. Experimental setting and analysis

### 4.1. Experimental data and setting

There is little research on relation extraction in the field of botany. First, data corresponding to the plant entity should be found. Based on climatic conditions and vegetation types, China's vegetation zones are divided into 13 regions, such as the warm temperate grassland and the temperate grassland [37]. Plants are selected from thirteen regions. Initially, the identified plant categories include evergreen trees, deciduous trees, evergreen shrubs, deciduous shrubs, vines, herbaceous plants, and bamboos [38]. Subsequently, plants related to these seven categories are chosen from the thirteen vegetation regions as data objects. The dataset used in this study is Landscape, which is scraped from websites such as the Flora of China. According to plant names, the dataset is crawled through parsing website URLs. And for unstructured data, regular expressions are used to match plant information. The next step is to identify the classification of relations. By consulting the classification information of plants from the Flora of China, the set of relations $R$ is artificially defined. R includes a total of 13 categories, such as aliases, cultural symbolism, and geographical distribution.

After defining the above architecture, the dataset is processed. Special characters and spaces are removed from the text, then the paragraphs in the text are split into words by the tokenizer. The regular expression (re) module in Python is used to remove characters like '&*' and the tokenizer can be jieba or other tools. Each word has a part of speech, making it easier to manually label triples. The labels "subject," "predicate," and "object" form a triple, where "predicate" is the relation. Triples stored in a dictionary, together with the data text, form the training dataset and testing dataset. The example of the data is shown in Fig 6:

where, the text as input is used to extract triples, the "triple_list" is the triple used for comparison with the predicted results. From Fig 6, Golden tree serves as a plant entity, having both the function of purifying the air and extracting essential oils. Multiple triples can be extracted from a sentence. The plant text dataset contains a total of 9,297 examples, with 5,578 examples in the training data, 1,896 examples in the validation data, and 1,823 examples in the test data. To effectively evaluate the performance of various relation extraction models, we use accuracy, precision, and recall as evaluation metrics.

In a sentence, there may be multiple triples. Sentences can be categorized into five cases according to the number of triples: $n = 1$, $n = 2$, $n = 3$, $n = 4$, $n \geq 5$. Additionally, there are four overlapping patterns, which are briefly described as follows.

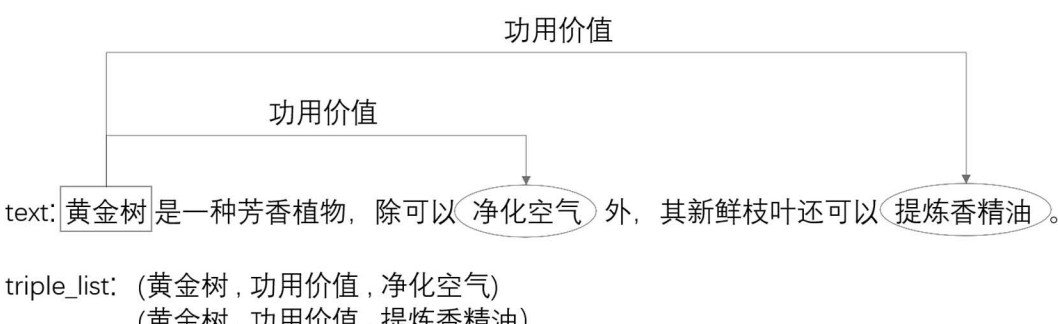

triple_list：（黄金树，功用价值，净化空气）
　　　　　　（黄金树，功用价值，提炼香精油）

**Fig 6. Examples of plant corpus.**

The first is normal overlapping pattern. The triples in the same sentence do not share entities and relations.

The second is single entity overlap (SEO). The triples in the same sentence share the same head entities or tail entities.

The third is entity pair overlap (EPO). The triples in the same sentence share the same entity pairs.

The fourth is subject object overlap (SOO). The triples in the same sentence overlap between head entities and tail entities. The head entities may include the tail entities of another triple, or the tail entities may include the head entities of another triple.

We take SEO as an example, highlighted in Fig 7. SEO shares entities, which can be either the head entities or the tail entities. In Fig 7, the triples share the head entities. Both "随意草" and "软枣猕猴桃" participate in the formation of several triples.

The model is evaluated from two perspectives: the number of triples and the overlapping patterns of triples. The evaluation metrics are accuracy, precision, and recall. This experiment is conducted using the PyTorch framework, encompassing the pre-training model BERT and other neural networks. The hardware configuration for this experiment consists of an AMD EPYC 7371 CPU with 28GB of operating memory, accompanied by an RTX A5000 GPU with 24GB of memory. As for the software configuration, the experiment utilizes the PyTorch 1.7.0 framework, Python 3.8 programming language, and is operated within the Ubuntu 18.04 environment. All key parameters within the experiment are set as follows: the maximum text length is restricted to 512, the adversarial training perturbation value is set at 0.005, the learning rate of the pre-trained model is 0.0001, and early stopping is implemented after 30 epochs. In the experiment, all compared models are based on the pre-trained model, BERT-base. Their hardware configuration, text length, and early stopping are also kept the same. Due to structural limitations, CasRel can only be set to a batch size of 1. The other models maintain the same batch size as our model.

1.软枣猕猴桃 可制果 药用 ，为强壮、解热及收敛剂；又是营养价值很高的 食品 。

1——< 软枣猕猴桃，功用价值，　　药用　　>
1——<软枣猕猴桃，功用价值，　　食品　　>

2.随意草 ：为 唇形科 随意草属 ，株高 60-120厘米 ，地上茎直立丛生，有根茎，多年生 草本 植物。

2——< 　随意草　，　高度　，60-120厘米 >
2——< 　随意草　，　属于　，　随意草属　>
2——< 　随意草　，　属于　，　唇形科　 >
2——< 　随意草　，植物分类，　草本　 >

**Fig 7. Examples of SEO pattern.**

## 4.2. Experiments and results

**4.2.1. Overall results.** Experiments results are presented in this section. To evaluate the extraction performance of our model, precision, recall, and F1-score are chosen as evaluation metrics. The following relational triples extraction models are used for comparison:

1. Tagging-based models

(1) A Relation-Specific Attention Network for Joint Entity and Relation Extraction (RSAN):

RSAN proposes a relation-based attention network, which utilizes a relation-aware attention mechanism to construct specific sentence representations for each relation, and then extracts the corresponding head and tail entities by sequence labeling [39].

(2) A Novel Cascade Binary Tagging Framework for Relational Triple Extraction (CasRel):

The article introduces a binary tagging sequence. First, CasRel identifies all possible head entities, and then it recognizes the tail entities related to the subject under the given relations [19].

(3) Potential Relation and Global Correspondence Based Joint Relational Triple Extraction (PRGC):

PRGC proposes a joint relational triple extraction framework based on potential relations and global responses [27].

(4) UniRel: Unified Representation and Interaction for Joint Relational Triple Extraction (UniRel):

UniRel proposes unified representations of entities and relations by jointly encoding them within a concatenated natural language sequence, which fully exploits their contextualized correlations [40].

2. Table-filling models

(1) Single-stage Joint Extraction of Entities and Relations Through Token Pair Linking (TPLinker):

TPLinker proposes a novel handshaking tagging scheme that aligns head entities and tail entities under each relation type [20].

(2) A Novel Global Feature-Oriented Relational Triple Extraction Model based on Table Filling (GRTE):

GRTE proposes a global feature-oriented table filling based RTE model that fill relation tables mainly based on two kinds of global associations [41].

(3) Joint Entity and Relation Extraction with One Module in One Step (OneRel):

A new relation-specific tagging strategy is proposed to determine the boundary tags of the head entity and the tail entity. OneRel uses a one-step approach to directly predict triples [21].

The results include the head entities, tail entities, and relations. Prediction is considered accurate only when each part of the results precisely matches the actual value. The extraction results of different models are shown in Table 3.

Our model outperforms other models in terms of precision, recall, and F1-score. Taking plant dataset as an example, the F1 score of our model obtains 6.9% improvements than CasRel, 3.2% improvements than OneRel, 2.9% improvements than TPLinker, 2% improvements than GRTE, 0.4% improvements than UniRel, and 1.4% improvements than PRGC. In training,

entities in CasRel model are real labels, but during inference, the prediction of tail entities depends on the identification of head entities. Errors in the head entities can lead to a decline in extraction accuracy. In comparison, the multi-task strategy of Bwdgv can enhance the interaction between subtasks, resulting in better extraction accuracy. TPLink model constructs three matrices to extract triples. But these matrices contain redundant information and have limited ability to generalize when extracting entities of varying lengths. GRTE improves global features based on TPLinker, but has low decoding and inference efficiency. Compared with these two, Bwdgv only constructs a matrix in entity pair matching, and optimizes entity recognition, enhancing its generalization ability. PRGC tends to recognize relations by remembering their positions rather than their semantics. In contrast to PRGC, Bwdgv enhances the semantic information of relations through multi-level information fusion. UniRel focuses on the integration of relational semantics, achieving mutual inference between entities and relations. In comparison, Bwdgv not only focuses on relations but also considers the balance between tasks. OneRel model employs a one-step decoding method to directly extract triples, but its design is more complex. Experimental results obtained on a single dataset may be accidental. To verify the model's performance in different data scenarios, we conduct experiments on Chinese Medicine Information Extraction dataset (CMeIE). The dataset contains two corpora: one is a pediatric training corpus, and the other is a common disease training corpus. It includes 14,339 training sentences, 4,482 test sentences, and 53 predefined relation types. Despite the increased complexity of the dataset causing a decline in the performance of all models, our model still surpasses other models in terms of precision and F1-score. This demonstrates the reliability of our model in different textual scenarios. Overall, the excellent performance of Bwdgv is mainly attributed to its optimized entity recognition strategy, enhanced semantic information processing capabilities, and efficient multi-task strategy. Meanwhile, the performance of Bwdgv in plant dataset during training is shown in Figs 8–10.

Initially, our model's performance on accuracy, precision, and F1-score is not satisfactory. But with training, our model's performance gradually surpasses other models. This indicates that our model effectively learns the textual information and captures the relation patterns between entities.

**4.2.2. Ablation study.** Additionally, to show the effectiveness of the improvements in our model, the results of ablation experiments are reported in Table 4.

First, we only keep the adjustment to word embedding component and observe the changes in model's performance. The accuracy, precision, and F1-score of the model still perform well. This demonstrates the effectiveness of the adjustment to the word embedding component. Perturbation introduced in the word embedding layer is constructed by adversarial training to reduce overfitting of tasks. When facing complex contextual information, this component enhances the model's generalization performance by encouraging the model's gradients to continually move beyond a certain range, thereby avoiding getting stuck in local optima. To measure the performance of the model under different perturbations, we adjust the value of $\Delta x$. At this point, the F1 scores of the models have all improved, indicating that introducing

**Table 3. Effectiveness of relational triple extraction for each model on datasets.**

| Models | plant dataset | | | CMeIE dataset | | |
|---|---|---|---|---|---|---|
| | prec | Rec | F1 | Prec | Rec | F1 |
| CasRel | 82.7 | 81.9 | 82.3 | 70.7 | 68.5 | 69.6 |
| TpLinker | 84.2 | 88.3 | 86.3 | 71.3 | 70.5 | 70.9 |
| PRGC | 87.0 | 88.6 | 87.8 | 72.5 | 70.0 | 71.2 |
| RSAN | — | — | — | 69.7 | 67.6 | 68.6 |
| OneRel | 87.3 | 84.7 | 86.0 | 72.8 | 71.5 | 72.1 |
| GRTE | 87.5 | 86.9 | 87.2 | 72.4 | 71.3 | 71.9 |
| UniRel | 88.2 | 89.3 | 88.8 | 72.9 | 71.6 | 72.3 |
| Bwdgv | 88.5 | 89.8 | 89.2 | 73.6 | 71.4 | 72.5 |

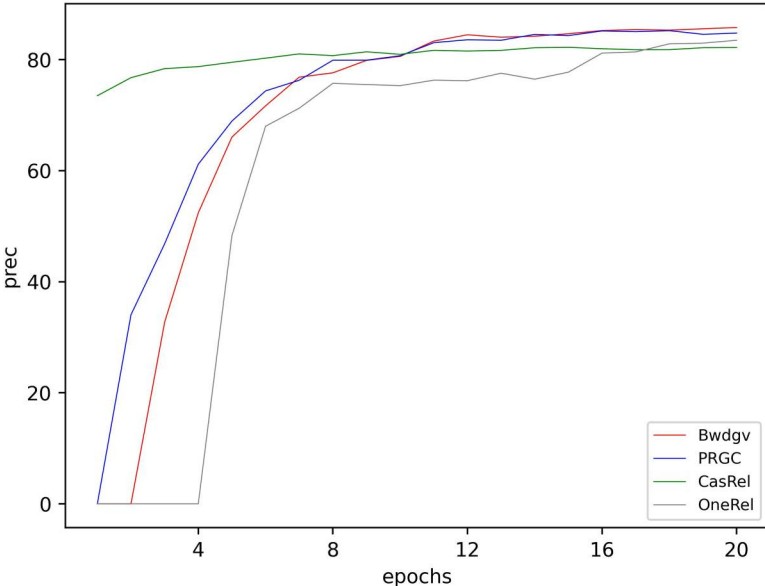

**Fig 8. Precision of different models in training.**

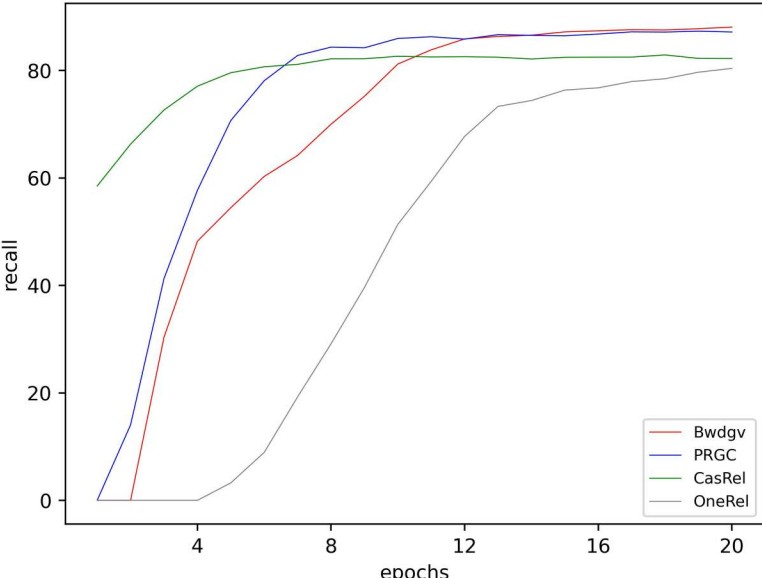

**Fig 9. Rec of different models in training.**

appropriate perturbations helps to enhance the model's performance. When $\Delta x$ is 0.5, the model performs better in terms of F1-score. When $\Delta x$ decreases to 0.005, Recall decreases from 89.8% to 89.3%, and Precision decreases from 88.3% to 88.1%, indicating that the perturbations have a more noticeable impact on the model's recall. Therefore, slightly larger perturbations can help the model identify more positive samples. Following, we only keep the optimization

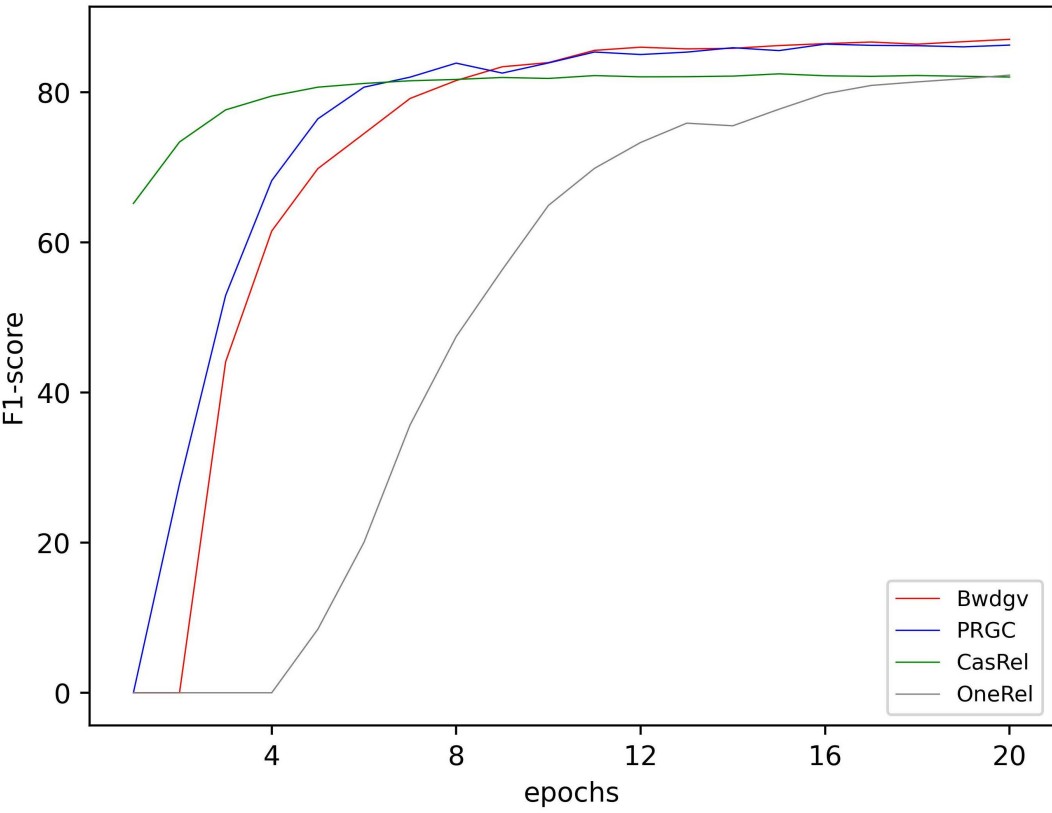

**Fig 10. F1-score of different models in training.**

**Table 4. Ablation study of Bwdgv.**

| | Models | prec | Rec | F1 |
|---|---|---|---|---|
| plant dataset | Bwdgv(adjustment to word embedding layer, $\Delta x = 0.005$) | 88.1 | 89.3 | 88.7 |
| | Bwdgv(adjustment to word embedding layer, $\Delta x = 0.05$) | 88.3 | 89.5 | 88.9 |
| | Bwdgv(adjustment to word embedding layer, $\Delta x = 0.5$) | 88.3 | 89.8 | 89.1 |
| | Bwdgv(optimization in relation prediction) | 87.8 | 89.2 | 88.5 |
| | Bwdgv(all, $\Delta x = 0.005$) | 88.5 | 89.8 | 89.2 |

component in Relation Prediction and the performance of the model still remains great. This indicates that it is important for the overall success of the model. This component enhances the accuracy of relation task by integrating multi-layer information, thereby effectively reducing the propagation of errors in downstream tasks. If the model relies solely on basic module of relation embeddings, which typically only captures shallow co-occurrence relations, it will be difficult to effectively identify and utilize context information at different levels, leading to a decline in performance in subsequent tasks. By introducing these two components, the performance of the model in multi-task learning has been enhanced.

**4.2.3. Detailed results on complex scenarios.** To evaluate the capability of our model in handling different overlapping patterns and sentences with different numbers of triples, we conduct further experiments on plant datasets. As shown in Fig 11, F1-score of four models is compared in all overlapping patterns.

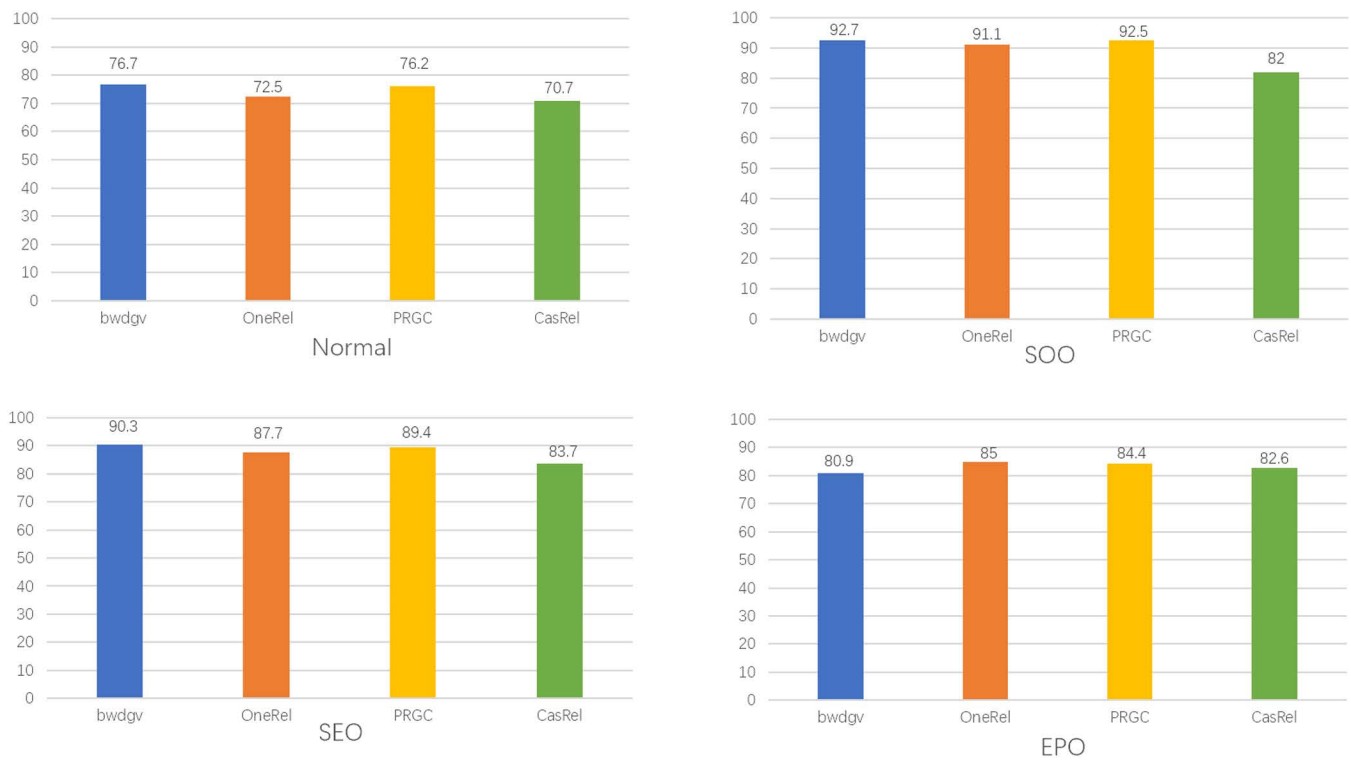

**Fig 11. F1-score of sentences with different overlapping patterns.**

Our model performs well on both SOO and SEO patterns. It achieves F1-score of 92.7% on SOO and F1-score of 90.3% on SEO. These patterns are related to entity sharing, so the identification of relations has little impact on the improvement of model performance. At this time, the performance of our model benefits from the adjustment to the word embedding. Since word embeddings are used for text representation, adding perturbations to them can enhance the robustness of entity recognition, allowing the model to better handle repeated entities. The performance of extracting triples in Normal and EPO patterns is not ideal, and except for the EPO pattern, our model outperforms other models. Thus, we analyze the proportion of overlapping triples in the dataset. The composition of overlapping triples is shown in Fig 12.

In the dataset, SEO pattern accounts for 83%, Normal pattern accounts for 10%, SOO pattern accounts for 5% and EPO pattern accounts for 2%. In SEO pattern, each plant has not only one attribute and classification. Most of the data is concentrated in the SEO pattern, which aligns with reality, so the model can learn more about the SEO pattern. In Normal pattern, entities are not shared, so each triple is independent and without correlation. But in other overlapping patterns, the triples are related. Adjusting the perturbation added to each individual triple is evidently more difficult than adjusting shared perturbation for several triples. Therefore, identifying the triples will be more challenging. The proportion of EPO pattern in the dataset is not high, which causes our model to perform poorly. Due to the limited amount of data related to EPO pattern, the noise introduced by perturbation makes it difficult for the model to learn useful information, resulting in a decline in extraction performance.

Next, we verify the model's performance under different numbers of triples. As shown in Table 5:

The number $N$ of triples in each sentence is not fixed. As $N$ increases, the F1-score of all models improves. When $N \geq 5$, the F1-score of our model is 91.5%, which is an improvement of 14.7% compared to when $N = 1$. The larger the value of N, the triples contained in the plant dataset become closer to the SEO pattern. A given sentence generally describes a plant, and as the number of extracted triples in a sentence increases, the probability of having the same head entity also

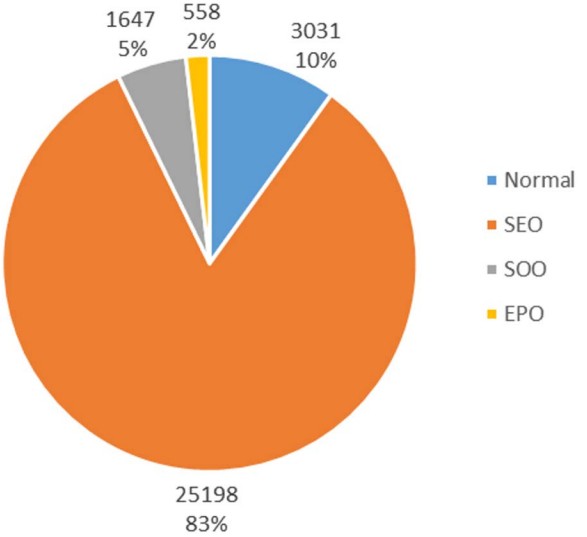

**Fig 12. Composition of different overlapping patterns in plant dataset.**

**Table 5. F1-score of models under different numbers of triples.**

| numbers of triples | Bwdgv | OneRel | PRGC | CasRel |
|---|---|---|---|---|
| N = 1 | 76.8 | 72.7 | 76.4 | 70.6 |
| N = 2 | 87.4 | 85.3 | 87.2 | 80.3 |
| N = 3 | 91.3 | 86.4 | 88.5 | 84 |
| N = 4 | 88.1 | 89.1 | 88.0 | 84.9 |
| N ≥ 5 | 91.5 | 88.7 | 91.0 | 84.9 |

increases. Thus, the model will learn more about SEO pattern. Benefiting from the perturbation in the embedding layer, our model has a stronger capability for processing information related to entity sharing. On the other hand, as $N$ increases, the number of relations also grows. When identifying relations, fusing multi-level information helps model capture contextual semantic information. Simultaneously, when predicting results, the attention mechanism is used to dynamically allocate resources to subtasks, balancing overall performance. Based on this, relations and entity pairs can be matched more accurately. As the number of triples increases, our model has an advantage in handling complex sentences.

**4.2.4. Instance analysis of triple extraction.** Through experiments and analysis, our model demonstrates great performance. For showing the effect of actual extraction, we extract triples from the following sentences. The extraction results are shown in Fig 13:

where 'pre' is the model's predicted result, 'True' is the actual result, the number is the relation and the blue marker is an incorrectly recognized or unrecognized mark. In this instance, the model can correctly identify the relations, but there are some errors in the recognition of entities. The extraction results of text 1 and text 4 lack some entities, which indicates that the model tends to miss certain elements when handling multiple attributes.

The prediction results of text 2 are consistent with the actual results, indicating that the model performs well with structured information. While in text 3, the boundaries of the entities are incorrectly identified, indicating that the model may make inconsistent errors in nuanced semantic comprehension. To enhance the model's memory, multiple attribute data samples can be added. Secondly, we can replace Bert with more powerful pre-trained models such as Roberta, or modify

```
1.匈牙利丁香害虫有毛虫、刺蛾、潜叶蛾及大胡蜂等。
pre:[ ('匈 牙 利 丁 香', '毛 虫', 9), ('匈 牙 利 丁 香', '大 胡 蜂', 9), ('匈 牙 利 丁 香', '刺 蛾', 9) ]
True:[ ('匈 牙 利 丁 香', '大 胡 蜂', 9), ('匈 牙 利 丁 香', '刺 蛾', 9), ('匈 牙 利 丁 香', '毛 虫', 9), ('匈 牙 利 丁 香', '潜 叶 蛾', 9) ]

2.鸡爪槭是槭树科, 槭属落叶小乔木; 树冠伞形; 树皮深灰色。
pre:[ ('鸡 爪 槭', '落 叶 小 乔 木', 6), ('鸡 爪 槭', '槭 属', 3), ('鸡 爪 槭', '伞 形', 4), ('鸡 爪 槭', '深 灰 色', 11), ('鸡 爪 槭', '槭 树 科', 3) ]
True:[ ('鸡 爪 槭', '落 叶 小 乔 木', 6), ('鸡 爪 槭', '槭 属', 3), ('鸡 爪 槭', '伞 形', 4), ('鸡 爪 槭', '深 灰 色', 11), ('鸡 爪 槭', '槭 树 科', 3) ]

3.牡丹花型宽厚, 被称为百花之王, 有圆满, 浓情, 富贵, 雍容华贵之意。
pre:[ ('牡 丹', '圆 满', 5), ('牡 丹', '富 贵', 5), ('牡 丹', '浓 情', , ', 5), ('牡 丹', '百 花 之 王', 5), ('牡 丹', '雍 容 华 贵 之 意', 5) ]
True:[ ('牡 丹', '圆 满', 5), ('牡 丹', '浓 情', 5), ('牡 丹', '富 贵', 5), ('牡 丹', '雍 容 华 贵', 5), ('牡 丹', '百 花 之 王', 5) ]

4.桧柏主要分布于我国西北天山、祁连山等干旱贫瘠环境中, 具有适应性强, 护坡固沙, 岸边防护, 城区净化空气等用途。
pre:[ ('桧 柏', '岸 边 防 护', 1), ('桧 柏', '净 化 空 气', 1), ('桧 柏', '护 坡 固 沙', 1) ]
True:[ ('桧 柏', '城 区 净 化 空 气', 1), ('桧 柏', '岸 边 防 护', 1), ('桧 柏', '西 北 天 山', 2), ('桧 柏', '护 坡 固 沙', 1), ('桧 柏', '祁 连 山', 2) ]

            "0": "别称"          "1": "功用价值"     "2": "地理分布"    "3": "属于"
            "4": "形状"          "5": "文化寓意"     "6": "植物分类"    "7": "物候期"
            "8": "生长习性"      "9": "病虫害"       "10": "胸径"       "11": "颜色"
            "12": "高度"
```

**Fig 13. Examples of different triple extraction results.**

the method of adding perturbations and the placement of the perturbations, to enhance the model's semantic comprehension. More importantly, entity recognition must also consider boundary issues. In the future, we can also consider first optimizing entity pairs and then matching relations.

### 4.3. Application of triples

Knowledge graph is a graphical structure model for representing and organizing knowledge. The triples extracted from the plant dataset can be used to construct a simple plant-related knowledge graph. To effectively construct a knowledge graph, triples are stored in a Neo4j database, where the entities in the triples serve as nodes and the relations between the entities serve as edges. In Fig 14, a knowledge graph is shown with information about some plants.

The circular node represents an entity, which can be either plant name or plant attribute; the color of the circular node indicates the relation between the entities. Circular nodes related to plant names serve as head entities and point to plant attributes with edges. In Fig 14, the information related to a specific plant can be quickly identified. For example, when searching for cedar, it can be found that its habit is sun-loving.

Constructing a knowledge graph through triples can provide technical support for quickly retrieving plant information. More importantly, knowledge graph can offer efficient services for subsequent applications such as plant knowledge Q&A. Relational triple extraction is the foundation for constructing knowledge graph. Therefore, its quality greatly affects the effectiveness of knowledge graph.

### 5. Conclusions and future work

In this paper, we explore joint extraction model based on plant datasets. Analyzing the text, 13 attribute categories are defined and plant entities are fixed as the head entities to narrow the scope of extraction. To alleviate the issues of error amplification and unstable parameter updates in multi-task parameter sharing, this paper proposes a model called Bwdgv. The model introduces perturbations in the word embedding layer and employs a multi-level information fusion strategy for relation prediction. We use accuracy, F1 score, and recall as metrics to evaluate model's performance. By comparing with

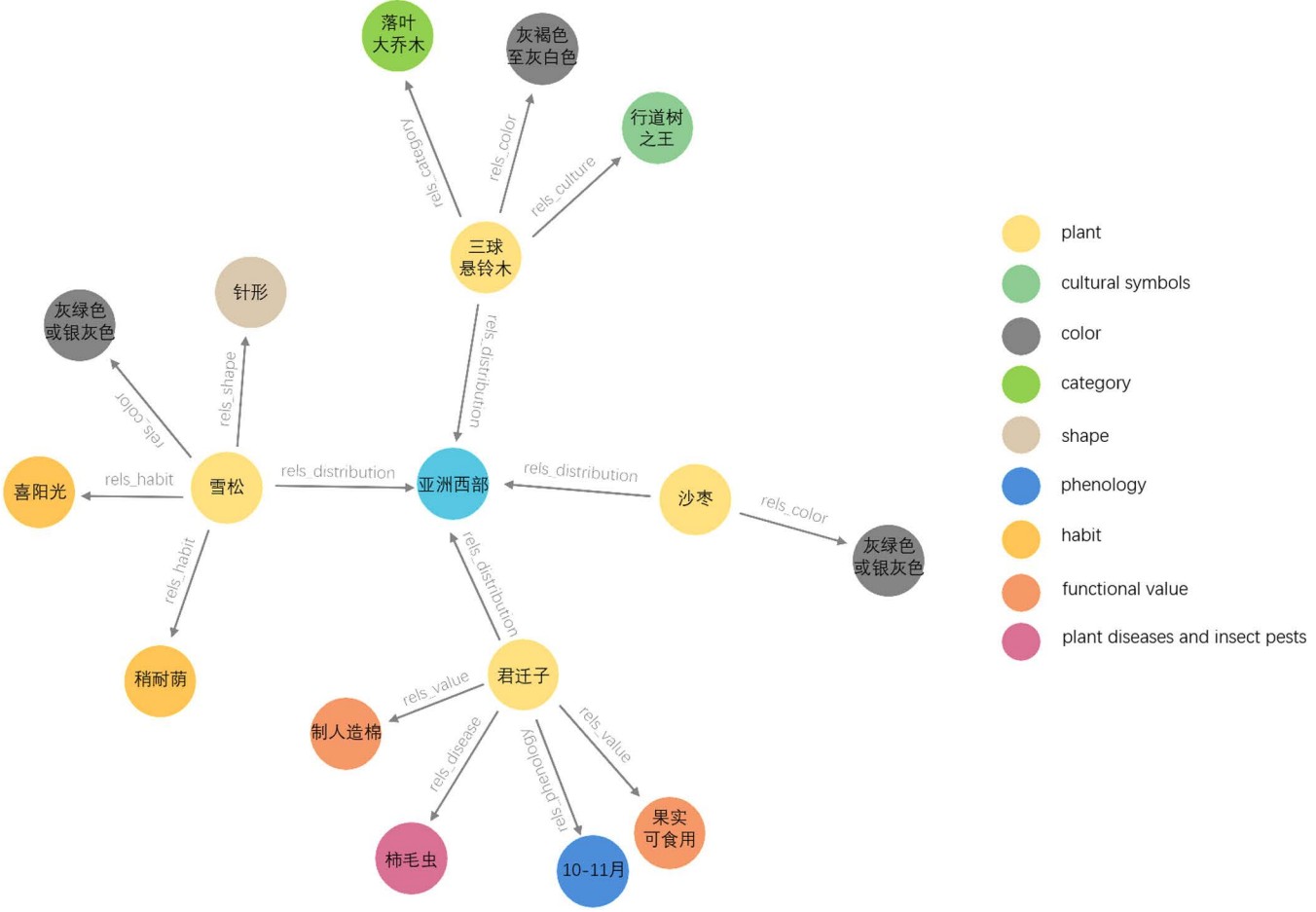

**Fig 14. Instance of plants knowledge graph based on relational triple extraction.**

other models such as CasRel, our model demonstrates superior extraction performance and excels in handling complex scenarios, such as overlapping patterns and sentences with different numbers of triples. In addition, based on the insights gained from this research, future attempts may involve improving entity relation extraction performance by incorporating additional semantic features in the relation extraction phase, optimizing perturbation for specific subtasks, or exploring alternative feature fusion methods in entity relation pair matching. Finally, these triples extracted from plant texts can be applied in fields such as knowledge graph to improve the rapid retrieval of plant information.

## Acknowledgments

The first author would like to express his gratitude to the mentors and friends for their help and encouragement during this project.

## Author contributions

**Conceptualization:** ZhiHao Zong.

**Formal analysis:** Hongtao Shan.

**Investigation:** Gaoyu Zhang.

**Project administration:** Gaoyu Zhang, Shuyi Zhang.

**Software:** ZhiHao Zong.

**Supervision:** Gaoyu Zhang, Shuyi Zhang.

**Writing – review & editing:** ZhiHao Zong, Gaoyu Zhang, George Xianzhi Yuan.

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
