## [Editor Report · Decision Letter 0]

PONE-D-24-32671

Plant Attribute Extraction Based on an Enhancing Three-Stage Model

PLOS ONE

Dear Dr. Zhang,

Thank you for submitting your manuscript to PLOS ONE. After careful consideration, we have decided that your manuscript does not meet our criteria for publication and must therefore be rejected.

I am sorry that we cannot be more positive on this occasion, but hope that you appreciate the reasons for this decision.

Kind regards,

Fu Lee Wang

Academic Editor

PLOS ONE

Additional Editor Comments:

This paper makes a fundamental mistake. It mentions that Chinese text is segmented by characters. In fact, Chinese text is segmented by words and Chinese segmentation is an important research topic. The author may submit the manuscript to some editor for Chinese language processing.

It is also mentioned that words in English can be split into smaller sub-words. This is not a common approach in English processing.

- - - - -

---

## [Author Response · Author response to Decision Letter 1]

15 Oct 2024

Our argument is backed by solid evidence and decent literature. The segmentation of characters is mentioned in several papers.

papers

Enhancing Pre-trained Chinese Character Representation with Word-aligned Attention

Character-based bilstm-crf incorporating pos and dictionaries for chinese opinion target extraction

Glyce: Glyph-vectors for chinese character representations

There are two ways to segment Chinese text. For example, the Chinese sentence: "野鸢尾是多年生草本植物". Segmented by words, "野鸢尾" will be token as a whole to form a vector. However, segmented by characters, "野鸢尾" will be segmented into "野"、"鸢"、"尾" . The individual character can form a vector respectively.

In fact, there is a confusing concept: a single Chinese character is also usually called a word, and a vector matrix is also called word embeddings. But for better differentiation and illustration, the concept of characters came later. In deep learning, the current foundational pre-trained model BERT is to cut Chinese into characters for vectorization. In the vocab.text which is a file in BERT, you can see that Chinese is stored by character. Thus, Chinese text can be segmented not only by words but also by characters.

In English, words can be split into smaller sub-words, which are often referred to as morphemes. A morpheme is an important concept in linguistics that refers to the smallest, meaningful unit of language. The basic model BERT uses sub-words as units. For example, the sentence "Iris dichotoma is a perennial herb." The English word 'dichotoma' can be divided into "di", "##ch", "##oto", and "##ma". In the vocab.text which is a file in BERT, you can see that English is stored by sub-words. It can be proved that English text split into sub-words is widely used in research.Here are some papers.

papers

Enriching Word Vectors with Sub-word Information

Morphology-based and sub-word language modeling for turkish speech recognition

Neural machine translation of rare words with sub-word units.

---

## [Decision Letter · Decision Letter 1]

Dear Dr. Zhang,

Thank you for submitting your manuscript to PLOS ONE. After careful consideration, we feel that it has merit but does not fully meet PLOS ONE’s publication criteria as it currently stands. Therefore, we invite you to submit a revised version of the manuscript that addresses the points raised during the review process.

We look forward to receiving your revised manuscript.

Kind regards,

Jin Liu

Academic Editor

PLOS ONE

Additional Editor Comments:

Based on the advice received, a revised version of this manuscript should be presented to address the points raised during the review process.

Reviewers' comments:

Reviewer's Responses to Questions

**Comments to the Author**

Reviewer #1: (No Response)

Reviewer #2: (No Response)

Reviewer #3: (No Response)

Reviewer #4: (No Response)

2. Is the manuscript technically sound, and do the data support the conclusions?

Reviewer #1: Yes

Reviewer #2: Yes

Reviewer #3: Yes

Reviewer #4: Yes

3. Has the statistical analysis been performed appropriately and rigorously?

Reviewer #1: Yes

Reviewer #2: Yes

Reviewer #3: Yes

Reviewer #4: Yes

4. Have the authors made all data underlying the findings in their manuscript fully available?

Reviewer #1: Yes

Reviewer #2: Yes

Reviewer #3: Yes

Reviewer #4: (No Response)

5. Is the manuscript presented in an intelligible fashion and written in standard English?

Reviewer #1: Yes

Reviewer #2: Yes

Reviewer #3: Yes

Reviewer #4: (No Response)

Reviewer #1: I would say this manuscript is publishable after major revisions. The reasons are as follows:

1.Formula labeling inconsistency. The third section contains inconsistent labeling of the same formula within the text, which may lead to reader confusion and negatively impact the article's readability.

2.Inadequate explanation of attention mechanism. While Section 3.4 mentions the attention mechanism, it lacks a detailed explanation of its specific implementation and how the computed weights influence the model's prediction process.

3.Limited dataset scale. The current dataset size is relatively small, with an insufficient number of samples, potentially compromising the model's generalization capability and the reliability of the experimental findings. It is strongly recommended that the authors expand the dataset and conduct additional experiments to validate the model's performance and robustness.

4.Suboptimal visual presentation. Several figures and tables are either excessively dense or poorly organized, diminishing their readability and overall visual effectiveness.

Reviewer #2: 1.It is recommended that the authors reorganize the content by dedicating a separate section to the ablation study. This would allow for a more systematic and in-depth analysis of each module's contribution to the overall model performance, thereby enhancing the clarity and scientific rigor of the manuscript.

2.The methodology section in Chapter 3 lacks comprehensive explanations for some formulas. Key variables and derivation processes are not adequately clarified. For instance, while Formula 5 describes the distance between vectors, it fails to elaborate on how this distance is utilized in adversarial training. Similarly, Formula 7 outlines the calculation of disturbances but omits the specific meaning and computational steps of the involved variables. Providing these details is crucial for readers to fully understand the technical foundations of the proposed approach.

3.Although the proposed model demonstrates certain advantages in the comparative experiments, the number of baseline models included in the comparison is relatively limited. Expanding the comparison to include more state-of-the-art models would strengthen the validity and persuasiveness of the experimental results.

4.Section 3.3 mentions the dimensions of the global matrix and the prediction method but does not provide a clear explanation of how the global matrix is constructed. A detailed description of this process is essential for reproducibility and for readers to fully grasp the methodology.

Reviewer #3: In the revised version, a lot of improvements have been made in the manuscript . However, a number of comments are shared which need to be addressed.

1) Title can be improved by sharing approach names like adding terms like deep learning etc for better understanding of the domain

2) “Framework of Model” heading does not convey enough information. Is it the proposed model?

3) Equation 5 presents the distance of the vector. However the equation is very generic and should be presented by changing symbols as per the domain in view on which the paper is written not generic symbols.

4) The related work section is poorly written and managed. Only 4 references are discussed and then whole framework of ETL is presented with detailed mathematical background. It is recommended that the related work should present the review of existing studies. Latest papers published in top journals should be discussed. Few are given for your guidance, these and more to be added. Also, try to add another heading and share ETL and equations and then equations of BiLSTM should be presented in that new heading.

Guo, X., Zhu, Y., Li, S., Wu, S., & Liu, S. (2025). Research and Implementation of Agronomic Entity and Attribute Extraction Based on Target Localization. Agronomy, 15(2).

Dalvi, P., Kalbande, D. R., Rathod, S. S., Dalvi, H., & Agarwal, A. (2024). Multi-Attribute Deep CNN-Based Approach for Detecting Medicinal Plants and their Use for Skin Diseases. IEEE Transactions on Artificial Intelligence.

Wang, M., Zhang, S., Li, R., & Zhao, Q. (2024). Unraveling the specialized metabolic pathways in medicinal plant genomes: A review. Front. Plant Sci, 15, 1459533. doi: https://doi.org/10.3389/fpls.2024.1459533

Chen, C., Wang, R., Chen, M., Zhao, J., Li, H., Ignatieva, M.,... Zhou, W. (2025). The post-effects of landscape practices on spontaneous plants in urban parks. Urban Forestry & Urban Greening, 107, 128744. doi: https://doi.org/10.1016/j.ufug.2025.128744

5) A table of symbols used in the paper be presented with description for better understanding.

6) Conclusions are presented however future research work should also be proposed in this section and the title of the section can be changed to Conclusion and Future work

Reviewer #4: 1、The three-stage joint extraction framework proposed in this article has achieved significant results in SEO overlap mode, but the discussion of existing models (such as ETL Span, OneRel, etc.) in the introduction and related work sections is relatively brief. Suggest adding comparisons with these latest models and clearly pointing out the technical and performance differences between our method and these models. This will help readers understand the innovation and advantages of the proposed method.

2、The integration logic of BERT wwm and adversarial training in Section 3.1 needs further clarification, especially on why perturbations are added at the word embedding layer instead of being processed at other layers, and how this design mitigates the negative impact of parameter sharing. In addition, it is recommended to further analyze the specific impact of this design on the stability of model training.

3、In the experimental section, it is necessary to supplement the comparison of the models in Table 2 and clarify whether the pre trained models used are consistent, as well as whether the hyperparameter settings are the same, to ensure the comparability of the experimental results. Meanwhile, the clarity of the experimental results and model architecture diagrams is poor. It is recommended to increase the resolution and annotation clarity of the charts to ensure their ease of interpretation.

4、Finally, it is recommended to quantify the specific impact of adversarial training on F1 score in the ablation experiment, in order to more clearly demonstrate the contribution of adversarial training to model performance and help readers understand its effectiveness.

**Do you want your identity to be public for this peer review?** For information about this choice, including consent withdrawal, please see our Privacy Policy

Reviewer #1: No

Reviewer #2: No

Reviewer #3: No

Reviewer #4: No

---

## [Author Response · Author response to Decision Letter 2]

13 May 2025

Reviewer #1: I would say this manuscript is publishable after major revisions. The reasons are as follows:

1). Formula labeling inconsistency. The third section contains inconsistent labeling of the same formula within the text, which may lead to reader confusion and negatively impact the article's readability.

Answer:

(We revised formula4 and the text�where g is the derivative of L(x+∆x, y;θ) at the word embeddings.

2). Inadequate explanation of attention mechanism. While Section 3.4 mentions the attention mechanism, it lacks a detailed explanation of its specific implementation and how the computed weights influence the model's prediction process.

Answer:

(We have added an explanation of the attention mechanism, and briefly explained how weights are assigned and the benefits of weight assignment.

The corrected part starts from line 488:

where the text vector and relation vector are directly concatenated to form a new composite vector. This operation allows the model to consider both the overall text information and specific relational information, thereby enhancing the accuracy of subject and object identification. Next, we use the attention mechanism to allocate weights to information, dynamically adjusting the model's focus on different pieces of information. The self-attention mechanism is adopted as the input consisted only of concatenated composite vector. For this composite vector, we use different linear transformations to generate the Query vector, Key vector, and Value vector. These transformations are usually fully connected layers, which map the input to different dimensional spaces. By computing the dot product between the Query vector and Key vector, a score matrix is obtained, reflecting the relevance of each element to all others. Then, to characterize the importance of the current information, the softmax function is used to convert the score matrix into a weight matrix. The weight matrix is finally used to perform a weighted calculation with the Value vector, resulting in the output vector for the position. After these steps, the text vector and the relation vector can be better integrated to highlight important information.

3). Limited dataset scale. The current dataset size is relatively small, with an insufficient number of samples, potentially compromising the model's generalization capability and the reliability of the experimental findings. It is strongly recommended that the authors expand the dataset and conduct additional experiments to validate the model's performance and robustness.

Answer:

(We acknowledge that the current dataset size is relatively limited, which is indeed a major limitation of this study and may affect the direct generalizability of the results. As you pointed out, accurately evaluating model performance typically requires a larger dataset. Regarding the dataset, we have conducted preliminary calculations. Since the dataset requires manual precise annotation, especially the marking method of entity relations needs meticulous judgment and verification. Based on our experience, approximately 30 pieces of data can be processed per day on average. This annotation process is not only time-consuming but also requires high professional knowledge and meticulousness from annotators to ensure the accuracy of entities and their relations. Considering the time required to revise the paper and my personal work needs, we apologize that we can only supplement a few hundred data points within a short period (e.g., one month). At the current stage, there are indeed significant time challenges in supplementing a large-scale dataset.

We fully understand your concerns about the potential impact of dataset size on the reliability of the evaluation. To present our work as rigorously as possible and enhance the credibility of the results, we have added the following content in the revised manuscript.

1. Added comparative experiments: We introduce two relevant advanced models (GRTE and UniRel) as new baselines for comparison to more comprehensively evaluate the effectiveness of our proposed method. The compared models are shown in table 3.

2. Added ablation study: For the key perturbation term Δx in the model, we conduct a more detailed quantitative analysis by setting different values of Δx to verify the changes in model performance, demonstrating the rationality of our method design. The modified part is shown in the part of 4.2.2. Ablation Study.

3. Added a description of the CMeIE dataset: The dataset contains 14,339 training data samples. Which is one of the most diverse and publicly available datasets, ensuring the reliability of the model’s results. The description is show in the end of the part 4.2.1. Overall Results.

We recognize that small-scale datasets indeed have inherent limitations. In specific research fields or exploratory work, researchers sometimes conduct preliminary validations under limited data scales to demonstrate the feasibility of new methods or ideas. For example, the commonly used dataset in SOTA, WebNLG, has a training data size of 5,019. Although the current scale of our dataset limits the direct generalization ability of the results, it provides a basis for us to verify the effectiveness of the model. We hope that the reviewers can understand the efforts we have made and the difficulties in supplementing large-scale datasets at the current time point. We will improve the dataset in subsequent research.)

4). Suboptimal visual presentation. Several figures and tables are either excessively dense or poorly organized, diminishing their readability and overall visual effectiveness.

Answer:

(

1. We merge table 2 and table 3 into one table for description, and rename it to table 3. This table only describes the comparison results of the experiment. The original table 3 is replaced by table 4, which only describes the ablation experiments. Separating the comparison experiments and ablation experiments, which enhances the readability of article.

2. We add description between Table 3 and Figure 8. Considering the conciseness of the article and that these formulas do not affect the description, we removed original formula 1, formula 2, and formula 5.

3. We adjust the position of Figure 12, moving it to Section 4.1 Experimental Data and Setting to enhance the readability of the summary in 4.2.3 regarding different triplet patterns. In the section 4.1, we provide an explanation for SEO.)

Reviewer #2:

1.It is recommended that the authors reorganize the content by dedicating a separate section to the ablation study. This would allow for a more systematic and in-depth analysis of each module's contribution to the overall model performance, thereby enhancing the clarity and scientific rigor of the manuscript.

Answer:

(We add a description of the comparison between different models and make the ablation experiments as a separate section. The corrected parts are in the section 4.2.1. Overall Results the section and the section 4.2.2. Ablation Study)

2.The methodology section in Chapter 3 lacks comprehensive explanations for some formulas. Key variables and derivation processes are not adequately clarified. For instance, while Formula 5 describes the distance between vectors, it fails to elaborate on how this distance is utilized in adversarial training. Similarly, Formula 7 outlines the calculation of disturbances but omits the specific meaning and computational steps of the involved variables. Providing these details is crucial for readers to fully understand the technical foundations of the proposed approach.

Answer:

(

d=√((x1-x2)^2+(y1-y2)^2+⋯+(n1-n2)^2 )

1.

The formula, is used to illustrate that after text is converted into vectors, the distance between two vectors is constant and it is inappropriate to add a perturbation Δx to this vector. Other methods should be considered. To prevent misunderstanding that this is the method adopted in this paper, the standalone formula should be removed. Thus, We explain what a one-hot vector is and emphasize that adding perturbations after converting text into vectors is ineffective.

The corrected part starts from line 354:

The vectors converted from text are essentially one-hot vectors. A one-hot vector has a value of 1 at only one position and 0 everywhere else. The distance between the any two vectors is always √2. Therefore, adding perturbations to text vectors also has no effect.

〖 min┬θ〗⁡〖E(x,y)~D[max┬(∆x∈Ω)⁡L(x+∆x,y;θ) ]〗 (3)

g=∇xL(x+∆x,y;θ)

∆x=ε g/〖||g||〗^2 (4)

2.

Formula (7) is defined below as Formula (4). The L function used in Formula (4) originates from Formula (3). We correct the formula and state that g is the gradient of the L function at the embedding. The L function can be understood as the entire operation process of the model. The parameters of the L function are explained in Formula (5). Finally, the chosen perturbation is the normalized value of g. The comprehensive improvement explains how to obtain the perturbation Δx and how to calculate the overall process.

The corrected part starts from line 381:

The core of adversarial training is to find an appropriate ∆x. Word embedding layer, as the direct interface between the model and input text, serve to map words into dense vectors. Adding perturbations here can directly affect the model’s understanding of vocabulary. In addition, compared to adding perturbations in deeper layers, perturbations in the word embedding layer are easier to control and do not have impact on the other structures of the model. Thus, we add perturbations in the word embedding layer. As shown in Fig. 3.

Figure 3. Word Embedding layer with perturbations.

The perturbation ∆x is directly added to the word embedding matrix. The input is first converted into a one-hot vector, and then transformed into a text vector through the word embedding layer. At this point, different samples will share the same perturbations, making the implementation easier. Finding the location where the perturbations need to be added, and then the perturbations can be calculated. The goal of perturbations ∆x is to maximize L(x+∆x,y;θ), and the direction of gradient ascent for L is exactly the direction in which L increases. We view the gradient of L to be proportional to ∆x. The calculation of the perturbations is shown in Equation (4)

g=∇xL(x+∆x,y;θ)

∆x=ε g/〖||g||〗^2 (4)

where g is the derivative of L(x+∆x,y;θ) at the word embeddings, ε is the proportional coefficient, normalization is used to prevent perturbations from becoming too large. When constructing perturbations, the first step is to compute the loss of forward propagation and record the gradients of backward propagation. At this point, gradients obtained from word embeddings are regularized to obtain perturbations. And then, the perturbations are added to the word embedding to construct a new input x+∆x. Finally, the loss of forward propagation and the gradient of backward propagation are calculated once again. It should be noted that in the first training, the gradient does not need to be updated. In the second training, the gradient to be updated is the sum of the gradients calculated in the first and second computations. At this time, the model learns the features of both normal samples and adversarial samples simultaneously, better adapting to data changes and reducing fluctuation in the training. When updating the gradient, the input of word embeddings should revert to the original x. As the update without considering disturbances, the model reduces its reliance on adversarial samples, ensuring the stability of parameters. Based on this, the model learns how to resist noise at the input level and ensures the stability of the training process.

)

3.Although the proposed model demonstrates certain advantages in the comparative experiments, the number of baseline models included in the comparison is relatively limited. Expanding the comparison to include more state-of-the-art models would strengthen the validity and persuasiveness of the experimental results.

Answer:

(We increased the number of models and conducted a comparative analysis. The corrected part is in the section 4.2.1 Overall Results.

The corrected part starts from line 643:

GRTE improves global features based on TPLinker, but has low decoding and inference efficiency. Compared with these two, Bwdgv only constructs a matrix in entity pair matching, and optimizes entity recognition, enhancing its generalization ability.

UniRel focuses on the integration of relational semantics, achieving mutual inference between entities and relations. In comparison, Bwdgv not only focuses on relations but also considers the balance between tasks.

4.Section 3.3 mentions the dimensions of the global matrix and the prediction method but does not provide a clear explanation of how the global matrix is constructed. A detailed description of this process is essential for reproducibility and for readers to fully grasp the methodology.

Answer:

(We were not clear enough in our description of the global matrix, and did not specifically indicate that the result calculated from formula (6) is the representation form of the global matrix. We have revised the narrative style and rewritten how to understand the global matrix and how to construct the global matrix.

The corrected part starts from line 459:

The extraction of plant entities and plant attributes along with the classification of plant attributes can be performed simultaneously. As shown in Fig. 1, we use a global matrix to predict plant entities(subjects) and plant attributes(objects) in the phase of entity pair extraction. The dimension of the global matrix is determined by the number of tokens in the sentence. If there are n tokens, the shape of the matrix is R^(n×n). The columns of the global matrix represent the starting positions of the subjects, and the rows represent the starting positions of the objects. The elements in the global matrix indicate whether the subjects and objects begin from these positions. To obtain the global matrix, the tokens of the subjects and objects are first concatenated vertically. Then, a fully connected layer is used to compute the concatenated information. The value of each element in the global matrix represents the confidence between a subject and an object. The higher the confidence, the more likely the begin position will be predicted. The extraction of plant entities and plant attributes remains a classification task, thus we use the sigmoid function to predict the begin positions of subjects and objects. The resulting prediction is the global matrix, which can be expressed as Equation (6).)

Reviewer #3:

Title can be improved by sharing approach names like adding terms like deep learning etc for better understanding of the domain

Answer:

(We have revised the title to make it more specific. The corrected title is: Plant Attribute Extraction: An Enhancing Three-Stage Deep Learning Model for Relational Triple Extraction.)

“Framework of Model” heading does not convey enough information. Is it the proposed model?

Answer:

(Yes, this heading refers to the proposed model. We change the heading “Framework of Model” to “The Framework of the Bwdgv Model”.)

3) Equation 5 presents the distance of the vector. However the equation is very generic and should be presented by changing symbols as per the domain in view on which the paper is written not generic symbols.

Answer:

( d=√(〖(x1-x2)〗^2+〖(y1-y2)〗^2+⋯+〖(n1-n2)〗^2 )。

This formula is used to illustrate the calculation of distances between one-hot vectors. However, this formula is general and relatively simple. Considering that this paper only explains that it is not suitable to add perturbations to vectors, and it is not the method of adding perturbations used in this paper, we remove this formula and d

---

## [Decision Letter · Decision Letter 2]

Plant Attribute Extraction: An Enhancing Three-Stage Deep Learning Model for Relational Triple Extraction

PONE-D-24-32671R2

Dear Dr. Zhang,

We’re pleased to inform you that your manuscript has been judged scientifically suitable for publication and will be formally accepted for publication once it meets all outstanding technical requirements.

Kind regards,

Jin Liu

Academic Editor

PLOS ONE

Additional Editor Comments (optional):

After carring out the reviewers suggestions, this manuscript can be accepted for publication now.

Reviewers' comments:

Reviewer's Responses to Questions

**Comments to the Author**

Reviewer #1: All comments have been addressed

Reviewer #4: (No Response)

2. Is the manuscript technically sound, and do the data support the conclusions?

Reviewer #1: Yes

Reviewer #4: Yes

3. Has the statistical analysis been performed appropriately and rigorously?

Reviewer #1: Yes

Reviewer #4: Yes

4. Have the authors made all data underlying the findings in their manuscript fully available?

Reviewer #1: Yes

Reviewer #4: Yes

5. Is the manuscript presented in an intelligible fashion and written in standard English?

Reviewer #1: Yes

Reviewer #4: Yes

Reviewer #1: This paper proposes a neuro-symbolic approach to visual dialogue that combines neural perception with symbolic reasoning. By introducing a conversation memory and procedural semantics, the method enables explainable and coherent multi-turn dialogue grounded in visual input.

In response to reviewer feedback, the authors have made the following updates:

1.The issue of inconsistent formula labeling has been resolved.

2.The explanation of the attention mechanism is now clear and sufficiently detailed.

3.The additional comparative experiments (GRTE, UniRel) and ablation studies have strengthened the empirical support, and the explanation regarding dataset scale is reasonable.

4.The layout and structure of tables and figures have been improved, significantly enhancing the readability of the manuscript.

Reviewer #4: This paper aims to transform text data into structured information, using a joint entity and relationship extraction method based on a tagging scheme, and proposing a three-stage "Bwdgv" model. By adjusting the word embedding layer and optimizing the relationship prediction mechanism of BERT, the model has improved its F1 score by 1.4% compared to the advanced PRGC method, and has good application prospects in knowledge graph construction and other applications. The author has made sufficient revisions to the article based on the review comments, and the overall quality has reached a publishable level. It is recommended to accept it.

**Do you want your identity to be public for this peer review?** For information about this choice, including consent withdrawal, please see our Privacy Policy

Reviewer #1: No

Reviewer #4: No

---

## [Editor Report · Acceptance letter]

PONE-D-24-32671R2

PLOS ONE

Dear Dr. Zhang,

I'm pleased to inform you that your manuscript has been deemed suitable for publication in PLOS ONE. Congratulations! Your manuscript is now being handed over to our production team.

Kind regards,

on behalf of

Professor Jin Liu

Academic Editor

PLOS ONE